# Phagotrophic Protists and Their Associates: Evidence for Preferential Grazing in an Abiotically Driven Soil Ecosystem

**DOI:** 10.3390/microorganisms9081555

**Published:** 2021-07-21

**Authors:** Andrew R. Thompson, Andrea J. Roth-Monzón, Zachary T. Aanderud, Byron J. Adams

**Affiliations:** 1Department of Biology, Brigham Young University, Provo, UT 84602, USA; rothmonzon@gmail.com (A.J.R.-M.); Byron_adams@byu.edu (B.J.A.); 2Department of Ecology and Evolutionary Biology, University of Connecticut, Mansfield, CT 06269, USA; 3Department of Plant and Wildlife Sciences, Brigham Young University, Provo, UT 84602, USA; zachary_aanderud@byu.edu; 4Monte L. Bean Life Science Museum, Brigham Young University, Provo, UT 84602, USA

**Keywords:** co-occurrence networks, *Sandona* sp., *Rhogostoma* sp., McMurdo Dry Valleys, Antarctica, variation partitioning, soil food webs

## Abstract

The complex relationship between ecosystem function and soil food web structure is governed by species interactions, many of which remain unmapped. Phagotrophic protists structure soil food webs by grazing the microbiome, yet their involvement in intraguild competition, susceptibility to predator diversity, and grazing preferences are only vaguely known. These species-dependent interactions are contextualized by adjacent biotic and abiotic processes, and thus obfuscated by typically high soil biodiversity. Such questions may be investigated in the McMurdo Dry Valleys (MDV) of Antarctica because the physical environment strongly filters biodiversity and simplifies the influence of abiotic factors. To detect the potential interactions in the MDV, we analyzed the co-occurrence among shotgun metagenome sequences for associations suggestive of intraguild competition, predation, and preferential grazing. In order to control for confounding abiotic drivers, we tested co-occurrence patterns against various climatic and edaphic factors. Non-random co-occurrence between phagotrophic protists and other soil fauna was biotically driven, but we found no support for competition or predation. However, protists predominately associated with Proteobacteria and avoided Actinobacteria, suggesting grazing preferences were modulated by bacterial cell-wall structure and growth rate. Our study provides a critical starting-point for mapping protist interactions in native soils and highlights key trends for future targeted molecular and culture-based approaches.

## 1. Introduction

Understanding the interactions between organisms in soil food webs is foundational to linking soil biodiversity and ecosystem functioning [1,2]. Phagotrophic protists are an integral component of virtually all soil communities and possess a diverse suite of cell morphologies and functional traits that allow species to consume a variety of prey from a range of soil pore sizes [3,4,5]. They are especially known for promoting bacterial diversity and mobilizing nutrients (as prey) to higher trophic tiers [6,7,8,9], as well as regulating the populations of fungi, some metazoa, and other protists through predation and competition [10,11,12]. Currently, soil phagotrophic protist functions are generalized from studies of in vitro systems involving only a few, easily cultured taxa and artificially depauperate microbial communities [13,14,15]. Characterizing the diversity of interactions that occur between specific phagotrophic protists and other organisms in natural soil communities is essential to more fully understand soil food web structure, biological controls on nutrient cycling, and soil ecosystem resistance and resilience [14,16,17,18]. However, soils are one of the most biodiverse habitats on Earth [19,20] and the corresponding number of interactions complicates efforts to disentangle relationships between individual taxa in situ [21,22]. Natural soil ecosystems with highly reduced biodiversity, like the ice-free McMurdo Dry Valleys (MDV) of Antarctica, are thus invaluable systems for investigating the relationships between soil protists and other biota [16,23,24].

Approximately 0.3% of the Antarctic continent is ice-free [25], and the largest expanse is found in the McMurdo Dry Valleys (MDV), on the border of the Ross Sea [26]. Here, extreme cold, ultraoligotrophic soils, short growing seasons, frequent freeze-thaw cycles, low soil moisture, high pH and high soil salinity contribute to an ecosystem devoid of vascular plants and dominated by microbiota [16,27,28,29,30]. Bacterivorous nematodes and rotifers are the dominant metazoans in the arid soils (<5% moisture) that make up 95% of the MDV landscape [16,31]. Phagotrophic protists are more taxonomically diverse and widespread than metazoans, fungi and non-vascular plants in MDV soils [23,24], but it is unclear how they interact with these and other soil organisms, such as bacteria, and contribute to the overall structure of the food web.

The MDV soil ecosystem is an established model for studying community assembly processes [16,32,33,34,35], yet current knowledge suggests that abiotic factors contribute more to structuring these communities than biotic [36,37]. However, analyses investigating the relative contributions of abiotic and biotic drivers to community structure have overlooked phagotrophic protists [36,37]. Protist grazers interact more directly with bacteria than metazoan grazers and tend to be more sensitive to changes in bacterial communities [38]. Therefore, phagotrophic protists may be structured more by biotic interactions (e.g., competition, predation, and preferential grazing) than abiotic variables compared to other MDV biota, but currently there is little direct evidence of these interactions in the MDV [23,39]. However, the high phylogenetic diversity of phagotrophic protists relative to other eukaryote taxonomic groups in this system [24] suggests a corresponding diversity of feeding- and motility-associated morphologies, and thus a potentially high functional diversity [15,18,40].

Competition between metazoans and protists for food in soils is generally assumed to occur but only sparsely documented [12,41]. Bacterivorous protists and metazoans can interact differently with bacterial communities due to differences in their relative sizes and feeding mechanisms. Protists tend to inhabit and graze in smaller pore sizes than metazoans, reproduce at rates more comparable to their bacterial prey, graze bacteria selectively and individually, and use enzymatic reactions to digest prey items [6,15,18,23,42,43,44]. Conversely, nematodes graze non-selectively [13,42] and use mechanical grinding to break up their food [43]. Some studies suggest that nematodes are more effective at digesting bacteria with thick cell walls (i.e., gram-positive) and that at least some protists (e.g., *Cercomonas* sp.) prefer bacteria with thin cell walls (i.e., gram-negative; [43,44,45]). However, both groups are highly diverse and often target the same resource (bacteria), so expecting competition between them is reasonable. In the MDV, metazoans, including nematodes, do co-occur with naked amoeba and flagellates [23], indicating that some degree of competition between these grazers may occur here.

If present, cytophagous protists (i.e., protists that feed on other protists [46]) probably regulate MDV protist communities more than metazoan predators, as the latter are more sparsely distributed [23,24,47,48]. Potentially cytophagous protists present in the MDV include the genera *Acanthamoeba* and *Pseudochilodonopsis* [24,46,49,50], yet no evidence of cytophagy by these species exists in this system. Instead, only the metazoan omnivore *Eudorylaimus* sp. (Nematoda) is known to be predatory, but it is usually restricted to moist soils adjacent to ephemeral streams or lake margins [16,48]. Also, members of the predatory tardigrade genus *Milnesium* frequent the ice-free regions in Antarctica but have so far not been found in the MDVs [51,52].

Most phagotrophic protists in the MDV are likely bacterivores, although fungivores, algavores, mixotrophs, potential predators and parasites are present [23,24,39]. Bacterivorous protists frequently have a species-specific impact on the composition and structure of bacterial communities [15,18,42] and grazing preference is regulated, in part, by bacterial traits, including cell wall thickness, growth rate, nutrient composition, and secondary metabolites [42,43,44,53]. The harsh physical environment of the MDV might benefit opportunistic grazers more than species-specialists [36,54]; however, the feeding behavior of different protists known to occur in the MDV are constrained by their distinct morphologies [42] and should give rise to some degree of trophic specialization even in this system. If preferential grazing does occur, the nutrient-limited soils and short timespans of hospitable conditions of the MDV [55,56] may select for protists that graze fast-growing bacteria capable of direct nutrient fixation from the atmosphere (e.g., carbon by phototrophs and nitrogen by diazotrophs).

To detect potential interactions involving phagotrophic protists in the MDV, we performed network analysis on co-occurrence patterns among bacterial and eukaryotic OTUs [4,57,58,59,60] derived from 18 spatially-sampled shotgun metagenomes that were reported on in a previous publication [24]. Network analysis determines whether co-occurrence between taxa is likely to be non-random (termed ‘association’) and provides a visualization of potential relationships between taxa in a dataset [57,58]. However, inferring biotic interactions from network associations requires care because of a variety of confounding factors, such as abiotic filtering, dispersal limitation, and the indirect interactions arising from a community of dynamic organisms [61,62,63]. To control for the effects of dispersal limitation and habitat suitability on co-occurrence patterns, we measured abiotic variables [24] and analyzed them against the associations in our networks using variation partitioning and MANOVA [59,60,64]. To improve the accuracy of our inferences, we compared the resulting associations with established relationships in the literature [4]. We hypothesized that: 1) The co-occurrence of biota better explain phagotrophic protist distribution in the MDV than abiotic drivers; 2) network analysis will reveal associations suggestive of intraguild competition and predation among metazoans and phagotrophic protists; and 3) associations between bacterivorous protists and bacteria will reflect a potential preference for fast growing, gram-negative, nitrogen- and carbon-rich bacteria rather than slow-growing, gram-positive, nutrient-limited taxa.

## 2. Methods

### 2.1. Shotgun Metagenome Sequencing and rDNA Identification

We used SSU sequences extracted from shotgun metagenomes and corresponding abiotic data to test distribution patterns between phagotrophic protists, other eukaryotes, and Bacteria for significant patterns of co-occurrence. Details concerning our sampling design, library preparation, sequencing, quality control, and bioinformatics were published previously [24]. Here, we provide a brief summary of our methods. Soil samples were collected from across the MDV over the span of several field seasons (2014–2017) and pooled to form 18 sites representing a wide variety of habitats, including several with visible organic matter (e.g., moss, biocrust). A shotgun metagenome library was prepared for each of the 18 sites by extracting DNA from the pooled samples using the DNeasy PowerSoil Kit (Qiagen). All 18 libraries were then sequenced over one and one-half Illumina Hi-Seq RapidRun lanes. Reads were trimmed and merged using Trimmomatic [65] and FLASH [66], respectively. The Metaxa2.2 default SILVA database (reference release 111) was used to extract and assign taxonomic statements (resolved to family) to bacterial SSU sequences [67]. Eukaryotic SSU sequences were found with the barrnap eukaryote hmm profile using nhmmer [68] and aligned against the PR^2^ database version 4.10.0 [69] and against the NCBI nt database release 230 with BLASTn v2.7.1+ [70,71]. Given that our shotgun-metagenome sequences did not necessarily overlap (as occurs in targeted amplicon sequencing) and could not be elongated by assembly due to low sequence abundance, OTUs could not be made using traditional clustering methods. Instead, OTU analogs were created by grouping sequences with the same assigned genus [72], a taxonomic level that avoids biases introduced with trying to identify species, while still enabling comparison of closely related or identical taxa across sites for the co-occurrence analysis. Although, such a conservative taxonomic resolution for bacteria (family) and eukaryotes (genus) may obscure potential interactions and the nuances of intra-genic and intra-family autecologies (e.g., see Glucksman, Bell, Griffiths and Bass [15]), species diversity is low in the MDV [16,48] and these designations may reflect actual species composition in this system more than in others. Moreover, our hypotheses are broad enough that we should be able to test them using even these broader taxonomic levels. Reads were normalized using the total counts method [73] to account for differences in sequencing depth between libraries [24]. Unassembled reads from each sample were uploaded to MG-RAST [74], under the project name SVLSoil18Proj090218. Broad functional categories (e.g., consumer) were assigned based on the literature [24,46].

Moisture, texture, pH, electrical conductivity (EC), total N, total P, total C, C:N ratio, N-NO_3_^−^ concentrations, distance to coast, elevation, and aspect were measured for each individual (not pooled) sample using standard soil analytic procedures [24]. We measured these attributes specifically because previous studies correlated soil moisture and pH to phagotrophic protist distribution in Antarctica [75,76] and distance to coast in the MDV is considered a strong predictor of biodiversity [36].The remaining variables are important drivers of the composition of MDV soil communities [35,47,55].

### 2.2. Constructing Association Networks

Associations were determined statistically by evaluating co-occurrence between every pair of OTUs against a randomized distribution model, implemented in the R package *cooccur* version 1.3 [57,58,77]. Using this method, pairs of OTUs that co-occur non-randomly (significance level of 5%) are termed associated. If pairs of associated OTUs co-occur significantly more frequently than predicted by the randomized distribution model, then the association is considered positive (i.e., the OTUs are aggregated). If the associated OTUs co-occur significantly less frequently than assumed by the randomized distribution model, then the association is considered negative (i.e., the OTUs are segregated; [57]). To test for associations, normalized counts for each OTU across all sites were converted to a presence-absence matrix. Pairs of OTUs were excluded from subsequent analyses when they were expected to share one site or fewer due to, either OTU in the pair occurring at low frequency in the dataset [77]. To visualize the resulting associations, individual co-occurrence networks were built for each of the five phagotrophic groups of interest (Cercozoa, Ciliophora, Amoebozoa, Discoba, and ‘other phagotrophic protists’) by converting OTU pairs into edge lists. The edge lists were then used to build unweighted unipartite networks using the R package *Igraph* version 1.2.5 [78].

### 2.3. Comparing the Influence of Abiotic and Biotic Factors on Protist Distribution

Partial redundancy analysis (pRDA), a type of linear model, was performed to quantify the relative contribution of abiotic and biotic factors (i.e., co-occurrence with other biota) in explaining the distribution of the five response groups: Cercozoa, Ciliophora, Amoebozoa, Discoba, and ‘other phagotrophic protists’ [64,79]. The pRDA partitions the total variation of the response variable, each of the five phagotrophic protist groups, into four fractions that correspond to: (1) Variation explained by only abiotic factors (the abiotic component), (2) variation explained by only biotic factors (the biotic component), (3) variation explained equally well by both abiotic and biotic factors (the shared component), and (4) unexplained variation (residuals). Each pRDA evaluated the normalized OTU abundance matrices of each response group against two explanatory variables, a biotic component and an abiotic component. The biotic component consisted of one of eleven biotic groupings (i.e., Ciliophora, Cercozoa, Amoebozoa, Discoba, Chlorophyta, Streptophyta, Bacteria, Fungi, Metazoa, Stramenopiles, ‘other phagotrophic protists’, and all biota combined) and the abiotic component consisted of all soil physiochemical and geographic variables discussed previously. To avoid circular testing of biotic variables, a biotic explanatory component was constructed for each pRDA by running a partial least squares regression between the five protist groups of interest (Ciliophora, Cercozoa, Amoebozoa, Discoba, and ‘other phagotrophic protists’) and the presence-absence matrix for each of the eleven individual biotic groups in the R package *pls* v.2.7.2. [80]. Either the first two components or those components that accounted for at least 50% of the co-variation were retained for each of the pRDAs [59,81].

A variety of measures were taken to detect and avoid collinearity among our abiotic variables. A Pearson correlation was used to test for all possible pairwise correlations and a variable was excluded from subsequent analysis if it had a correlation above 0.70 (the variables “C:N ratio” and “Distance to Coast” were excluded in this way). Each variable’s inflation factor (VIF) was checked after the pRDA was performed and it was determined that all abiotic variables were acceptable (below 10; [79]). The significance of variation was tested through a permutation test (999 permutations) using the *vegan* package version 2.5.6 in R [82]. To prevent inflation of R^2^ values, the variation explained by each fraction is reported as the adjusted coefficient of multiple determination (R^2^_adj_, [64]).

The results of the variation partitioning are useful to explore the impact of abiotic drivers at a broader taxonomic level (e.g., phylum). However, to evaluate the influence of key abiotic processes (e.g., dispersal and habitat filtering) at higher taxonomic resolution, a series of MANOVAs were performed on individual associations in our networks, following previous studies [60,83]. Briefly, associated pairs of OTUs were assigned OTU co-occurrence states for each site sampled (i.e., 11 if co-present, 00 if co-absent, and 01 or 10 if only one taxon in the pair was present). Then, the OTU states for each pair (independent variable) were run against geographic coordinates and abiotic variables (dependent variables) in separate series of MANOVA. A principal component analysis was used to include as much abiotic variation in as few variables as possible for the MANOVAs. We checked whether data for each dependent variable passed test assumptions only after the dataset was partitioned according to species pairs and OTU states and analyzed only those taxon pairs for which the partitioned data did not violate any assumptions. Testing assumptions only after partitioning data into individual pairs is a more conservative approach than the testing carried out previously, but we feel the sensitivity of the results to ambiguity warrants additional caution. All analyses were run in the R environment version 3.6.3, unless otherwise specified [84].

### 2.4. Inferring Biological Significance from Network Associations

After controlling for abiotic factors, the literature was used to infer potential biological significance from network associations. Here, we assumed a negative association between two taxa with similar functional roles (e.g., bacterivore) suggested competitive exclusion [61]. If a positive association occurs between a known or probable predator or grazer and an appropriate prey organism, then we inferred a trophic relationship [61]. However, predation is frequently opportunistic, and we assume this to be especially true in the low energy environment of the MDV. Therefore, if a potential predator exists in the dataset but was not associated with an appropriate prey item in the association network, then either (1) it does not operate as a predator in this system, (2) is an opportunistic or a trophic generalist, or (3) its usual prey was not included in our networks.

Associations between phagotrophic protists and bacteria in the network analyses indicated potential feeding interactions and enabled the testing of our hypothesis that bacterivorous protists graze MDV bacteria preferentially. To qualify associations in terms of these parameters, we constructed a palatability chart by comparing a palatability index developed herein against the growth rate of each bacterial OTU in association with a phagotrophic protist in the co-occurrence networks. To generate the palatability index, each bacterial OTU was scored based on three traits: Cell wall thickness (i.e., gram-negative or gram-positive), ability to fix atmospheric nitrogen, and ability to fix carbon. Although other traits exist that mediate protist prey preference, e.g., secondary metabolites [53], such traits are not characterized broadly enough nor are they conserved at higher taxonomic levels to make their inclusion in this analysis feasible. A bacterial OTU received one point for each trait that increased its palatability (i.e., gram-negative, nitrogen fixing, or photosynthetic) and received no points for a trait that did not increase its palatability (i.e., gram-positive, incapable of fixing nitrogen, or incapable of fixing carbon). The highest palatability score, a three, would thus be given to a gram-negative, nitrogen-fixing phototroph and the lowest possible score would be given to a gram-positive heterotroph incapable of fixing nitrogen. Trait designations for certain taxa were estimated by phylogenetic association [85,86] because (1) the ability to fix nitrogen and photosynthesize are not known for all bacteria, and (2) because our bacterial OTUs were only resolved to family.

rRNA copy number was used as a proxy for growth rate [87] and was estimated using The Ribosomal RNA Database (*rrn*DB), which provides copy number counts at the genus level [88]. Since bacterial OTUs in our dataset were resolved only to family, the rRNA copy number for each OTU was estimated using the mean and median rRNA copy numbers for all genera in each family. We excluded a bacterial OTU from the palatability chart if (1) the OTU did not exist on the rRNA database, or (2) its taxonomic resolution in our dataset was higher than family (e.g., OTUs marked as unidentified orders or classes or *incertae s**edis*). Approximately half of the bacterial OTUs (34 of 74) were thus removed. Each of the remaining bacterial OTUs were plotted according to their palatability (*y*-axis) and growth rate (*x*-axis). Protist consumers were then plotted next to each bacterial family based on the positive or negative associations in the co-occurrence network analyses. In order to determine whether the associations between bacteria and phagotrophic protists were significantly correlated with the palatability score or growth rate, we performed a Kendall’s tau-b correlation using the richness of associated protist taxa at the genus and phylum level as the dependent variables.

## 3. Results

### 3.1. Biotic vs. Abiotic Drivers

Biotic factors (i.e., co-occurrence with other biota) appear to be more important for explaining the observed distribution of all five phagotrophic protist groups than abiotic factors (soil physiochemistry and geography; Figure 1). When the biotic component consisted of all individual biotic groups combined (i.e., ‘all biota’), neither the abiotic nor the biotic components alone explained the distribution pattern of any phagotrophic protist response group with high confidence (*p* < 0.05; Appendix A). Moreover, the abiotic and the shared (i.e., patterns in the distributions that are explained by either biotic or abiotic factors) components together were likewise not significant in explaining distribution for any response group. Instead, the biotic and the shared components together explained at least 27% of the distribution of every phagotrophic protist group with high confidence (pRDAs for Cercozoa: R^2^_adj_ = 0.32, *F*_(5,15)_ = 2.52, *p* = 0.003; Ciliophora: R^2^_adj_ = 0.28, *F*_(5,15)_ = 2.23, *p* = 0.001; Amoebozoa: R^2^_adj_ = 0.61, *F*_(5,15)_ = 6.04, *p* = 0.001; Discoba: R^2^_adj_ = 0.76, *F*_(5,15)_ = 10.93, *p* = 0.001; ‘other phagotrophic protists’: R^2^_adj_ = 0.98, *F*_(5,15)_ = 184.99, *p* = 0.001). Overall, all measured abiotic and biotic variables accounted for a large proportion of the distribution, though Amoebozoa had more unexplained distribution (residuals) than Ciliophora, Cercozoa, Discoba, or ‘other phagotrophic protists’ (Cercozoa: R^2^_adj_ = 0.30; Ciliophora: R^2^_adj_ = 0.42; Amoebozoa: R^2^_adj_ = 1.1; Discoba: R^2^_adj_ = 0.23; ‘other phagotrophic protists’: R^2^_adj_ = 0.01). When considering individual taxonomic subgroups (e.g., Cercozoa, Ciliophora, Bacteria, or Streptophyta) instead of ‘all biota’ as the explanatory variable, the abiotic, biotic, ‘abiotic and shared’, and ‘biotic and shared’ components were significant in 0 of 50 tests, 6 of 50 tests, 10 of 50 tests, and 39, of 50 tests, respectively (5 response groups and 10 explanatory biotic subdivisions makes a total of 50 tests; Appendix A).

### 3.2. Trends in Network Analyses

Out of 90 phagotrophic protist OTUs in the metagenome dataset, 26 formed associations (i.e., a non-random co-occurrence pattern) with other OTUs for a total of 167 associations, 132 aggregates (positive associations) and 35 segregates (negative associations; Figure 2, Appendix A). The majority of associations involving phagotrophic protists were aggregates (79%; Appendix A) and all segregate associations in all protist networks were with bacteria (Figure 2). Most of the associations were with Cercozoa (57%), while Discoba had the fewest (0.06%). Ciliophora had the highest proportion of segregate associations of any response group (47%) while ‘other phagotrophic protists’ had the smallest proportion of segregate associations (6.25%). When controlling for the effect of abiotic filtering and dispersal limitation on associations, only four aggregate and zero segregate associations possessed sufficient data to pass the assumption of multivariate normality for the MANOVA (Appendix A). All four aggregate associations were found unlikely to be associated due to abiotic filtering or dispersal limitation (Appendix A) and were considered to be associated due to biotic processes.

### 3.3. Associations with Eukaryotes

All associations between phagotrophic protists and other eukaryotes (including other protists) were aggregate and comprised a small proportion (23%) of total associations. Three associations were with other phagotrophic protists, 14 with metazoans, one with fungi, 17 with streptophytes, and two with chlorophytes (Figure 2; see Appendix A for OTU names). Among those associations occurring between consumers (i.e., phagotrophic protists and metazoans), all were aggregate but we could not test whether the associations were due to abiotic filtering or dispersal limitation as none of them passed the assumption of normality for MANOVA (Appendix A). A single Cercozoan taxon (*Sandona*, OTU 257) was associated with a disproportionate number of taxa relative to the total number of associations between phagotrophic protists and other eukaryotes (25%). *Sandona* also associated with the greatest diversity of taxa of any OTU in the network (two metazoans, one amoebozoan, one ‘other phagotrophic protist’, one fungus, seven streptophytes, and two chlorophytes) and was the only protist to associate with a fungus. *Protocanthamoeba* (OTU 45) and *Acanthamoeba* (OTU 44) possessed the most associations among the Amoebozoa (67% combined), an unidentified Oxytrichidae (OTU 33) possessed the most associations among the Ciliophora (46%), and only a single OTU formed associations for both Discoba (*Keelungia*, OTU 134) and ‘other phagotrophic protists’ (the Stramenopile *Spumella*, OTU 275). The only nematode (*Plectus,* OTU 235) and tardigrades (OTU 235 and OTU 238) recovered appeared only in the Cercozoan network. Rotifers (e.g., *Adineta*, OTU 224) were the only metazoans to appear outside of the Cercozoa network and associated with a taxon from each phagotrophic response group except Discoba. Streptophyte OTUs 101 and 117 (*Bryum* and *Lygodium*) appeared in every phagotrophic protist network except for Discoba (Figure 2A) while Chlorophyta (OTUs 82 and 90) and the only fungal taxon (*Sporolobomyces*, OTU 188) associated exclusively with *Sandona*.

### 3.4. Associations with Bacteria

The majority of phagotrophic protist associations were with bacteria (77%) and most (73%) were aggregate. Bacteria in associations with phagotrophic protists belonged to 14 phyla, but the majority (67%) were from the following phyla: Proteobacteria (38%), Actinobacteria (14%), Chloroflexi (8%), and Acidobacteria (7%; Figure 3). Most (54%) phagotrophic protist OTUs were associated with at least one proteobacterium and all such associations were aggregate except for those between proteobacteria and the ciliophoran *Pseudochilodonopsis* (OTU 23; Figure 2B). Actinobacteria taxa segregated from phagotrophic protists the most (12% of all associations with bacteria) and accounted for nearly half (42%) of all segregate associations in the networks (Figure 2; Appendix A). Three of 81 bacterial families associated with at least one OTU from each protist response group: Rhizobiaceae (Proteobacteria, OTU 501), Trueperaceae (Deinococcus-Thermus, OTU 545), and group wr0007 in the Rhodospirillales (Proteobacteria, OTU 642; Appendix A). Rhizobiaceae always aggregated with phagotrophic protists, and Trueperaceae (OTU 545) was always segregate.

After scoring the palatability of bacterial families associated with phagotrophic protists, half (20 of 40 families) scored 1 out of 3 possible points (Appendix A). No bacterial family scored 3, five scored 2, and fifteen scored 0. Most families were gram-negative (25 of 40) while 3 and 2 families were estimated to include diazotrophs or phototrophs, respectively. All bacteria that scored 0 were gram-positive, all that scored 1 were gram-negatives that were not diazotrophs or phototrophs, and all that scored 2 were gram-negatives that were either diazotrophs or phototrophs (3, and 2 bacterial families, respectively; Appendix A). rRNA copy numbers ranged from 1 to 9, although most (90%) fell between 1 and 5 (Appendix A). The bacteria with the highest copy number (9) possessed the lowest palatability score (0; Figure 4; see Appendix A for a version of the chart that includes OTU names). Bacterial families that were negatively associated with a phagotrophic protist taxon cluster in the bottom left quadrant of the chart where rRNA copy number and palatability score are lower. Conversely, the diversity of associated phagotrophic protist groups increases as rRNA copy number and palatability score increase (Figure 4 and Appendix A). Statistical analyses showed some evidence for a negative correlation between protist richness and rRNA copy number for negative associations, but the relationship was only nearly significant (Phylum: τ_b_ = −0.362, *p* = 0.071; Genus: τ_b_ = −0.376, *p* = 0.058) (Appendix A). The OTU *Sandona* was associated with bacteria at all palatability scores and across a broad range of rRNA copy numbers (i.e., 2–9). Conversely, Ud. *Rhogostoma*-lineage does not occur in the highest palatability score in the chart (2) and does not associate with a family that has an rRNA copy number greater than four (Figure 4).

## 4. Discussion

We explored the potential interactions that may occur between phagotrophic protists and the rest of the soil microbiome in a model soil ecosystem using metagenomic datasets and co-occurrence analyses. We first tested whether the co-occurrence of biota or abiotic factors was a more appropriate explanatory variable for phagotrophic protist distribution in the MDV. Our results indicate that MDV phagotrophic protists are not primarily structured by abiotic processes (Figure 1), leading us to accept our first hypothesis. In addition, we found that most co-occurrence among MDV taxa is random (Appendix A), which supports the paradigm that MDV soils host food webs with few species-specific interactions and a relatively low degree of trophic specificity [36,54]. We next predicted that we would find associations suggestive of intraguild competition and predation. However, we found little indication for either among non-randomly co-occurring taxon pairs and thus reject our second hypothesis. Finally, we hypothesized that interactions between bacterivore protists and bacterial taxa would reflect preferential feeding. Beyond recovering numerous non-random associations between protist and bacterial taxa, we found that associations between protists and bacteria appear to correlate with prey palatability, especially cell wall structure and growth rate, and thus accept our final hypothesis. Controlling for dispersal limitation and abiotic filtering supported our findings that at least some phagotrophic protists associate with MDV bacteria preferentially, notwithstanding our conservative assumption testing that resulted in few analyzed species pairs. Future studies should strive to incorporate as broad a sampling design as possible to ensure they are able to control for these confounding factors without violating important assumptions.

A lack of associations suggesting competition between Metazoa and protists is not unreasonable as these two groups often occupy different soil pore size ranges and graze prey with different levels of selectivity [13,42,43]. However, a lack of evidence for competition among protist taxa was unexpected but may reflect the presence of spatially overlapping but temporally distinct soil communities, each regulated by a different set of protist consumers. The distinguishing factor between these communities might be microhabitat preferences (e.g., differences in optimal growth temperatures among grazers and prey) or differential prey preference [89].

The only potentially omnivorous taxon to associate with other eukaryote consumers was the amoebozoan *Acanthamoeba* (Figure 2C, OTU 44), a genus that contains species that consume a variety of non-bacterial prey [10,49]. The MDV’s only known predator, the omnivorous nematode *Eudorylaimus*, was absent from the networks but present in the dataset, albeit in very low abundance [24]. As the abundance and distribution of each of these potential predators is low [24,90], detecting any relationship between them and (other) phagotrophic protists will require focused study in sites where these organisms co-occur.

Phagotrophic protists in the MDV appear to prefer more palatable (score > 0) and faster growing (rRNA copy number > 4) bacterial prey, but not all protists avoided less palatable prey (Figure 4). In particular, ciliates aggregate only with bacteria with high palatability scores and growth rates, potentially because ciliates require energy-dense food to fuel their energetically demanding motility [91]. The bacterial phyla that phagotrophic protists aggregated and segregated with most frequently were the gram-negative Proteobacteria, and the gram-positive Actinobacteria, respectively (Figure 3). Other studies suggest that at least some bacterivorous protists prefer gram-negative Proteobacteria over gram-positive Actinobacteria [38,43,44]. To our knowledge, this is the first evidence of this phenomenon at a community level in natural soils.

Previously, the testate amoeba *Rhogostoma* (Cercozoa, Figure 2A, OTU 259) and an unidentified taxon in the family Sandonidae (Cercozoa, Figure 2A, OTU 258) were highlighted for their widespread distribution and high abundance relative to other protist genera (Thompson et al. 2020). We found that neither taxon accounted for a large proportion of total associations (7.5%, and 5.4% respectively). Instead, the Cercozoan *Sandona* (Figure 2A, OTU 257) associated with the greatest number (42 or 25% of the total) and diversity of taxa. This apparent incongruency may indicate that *Rhogostoma* and the unidentified Sandonidae are habitat and/or trophic generalists or that their greater distribution increases the resolution and robustness of their associations, filtering out false positive. Unlike *Rhogostoma* and the unidentified Sandonidae, which were each distributed across moist and arid sites, *Sandona* was only recovered from sites with > 5% soil moisture and may be an integral member of and a potential hub of trophic energy flow in the moist soil community [17]. This interpretation is strengthened by the taxon’s dominance in soils worldwide [92,93] and its tendency towards species-specific grazing [15].

Our study demonstrated that the MDV may host a food web with more species-specific interactions than previously thought, underscoring the utility of this system for exploring the roles protists and other members of the soil microbiome play in soil food webs. Co-occurrence network analysis, despite some of its limitations [60,62,63], may thus be essential for resolving taxon-specific interactions in soil microbial communities where in-depth functional studies (e.g., culture-based) are technically and logistically prohibitive. This is especially true in systems (like the MDV) where limited biodiversity and strong abiotic gradients can make interactions more detectable. Future research should verify the nature of the associations highlighted in this paper, using in vitro studies; higher resolution molecular work (e.g., primer amplicon studies and deeper metagenome sequencing) that can improve taxonomic resolution from family and genus to species or even populations; controls for small-scale abiotic and spatial filtering effects (e.g., soil heterogeneity within a sample site and vertical soil structure); and culture-based assessments of environmental tolerances and feeding preferences [56,94,95]. Achieving these goals will lay the groundwork for characterizing the links between ecosystem functioning and biodiversity for an entire microbial soil community [16], and will facilitate an improved understanding of ecological processes within and beyond the MDV system.

## Figures and Tables

**Figure 1 microorganisms-09-01555-f001:**
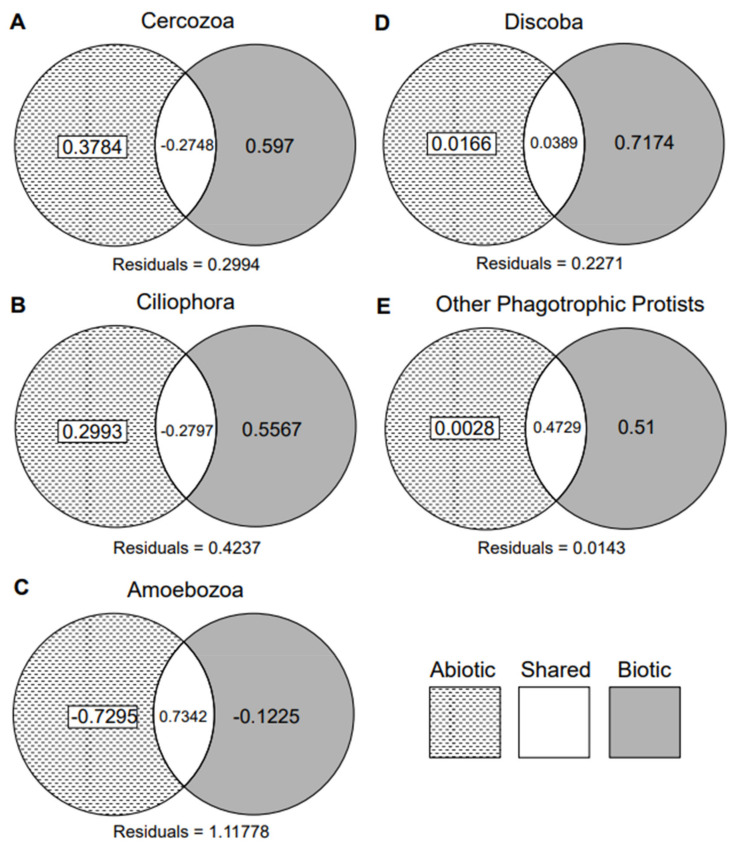
Phagotrophic protists are structured more by biotic factors rather than abiotic factors. Variation partitioning for five phagotrophic protist groups with all biota as biotic component, and environmental variables as abiotic component. Adjusted R^2^ values are shown inside each Venn diagram component (abiotic, shared, and biotic) and for the residual. Variation partitioning for (**A**) Cercozoa, (**B**) Ciliophora, (**C**) Amoebozoa, (**D**) Discoba, and (**E**) other phagotrophic protists.

**Figure 2 microorganisms-09-01555-f002:**
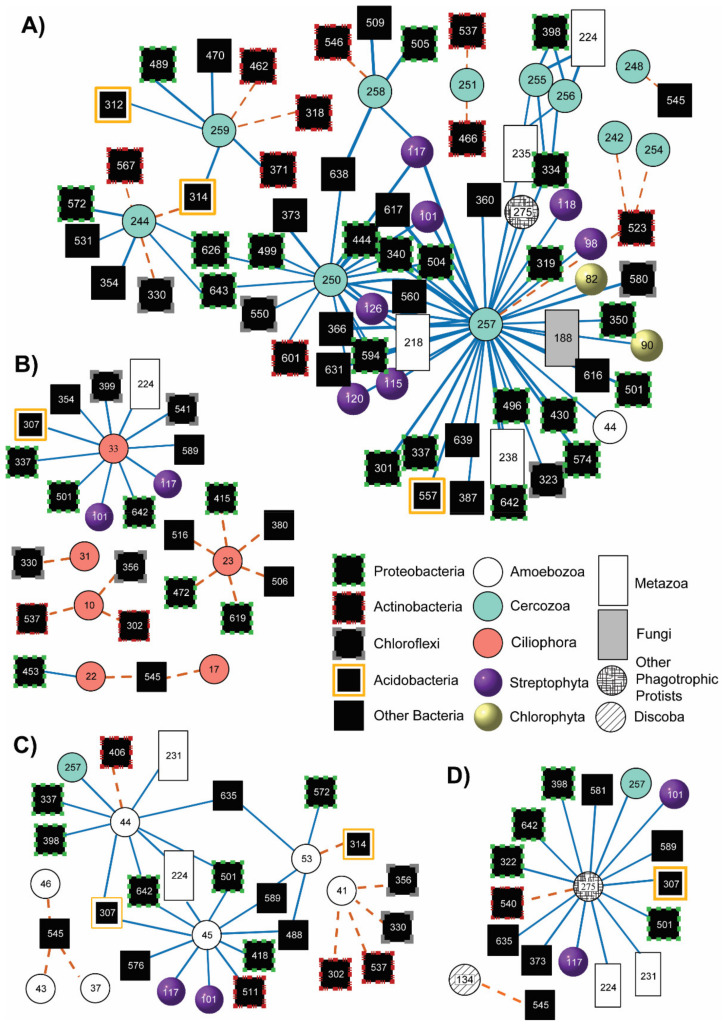
(**A**–**D**)—Networks suggest a complex food web structure. Visualizations of pairwise co-occurrence associations for (**A**) Cercozoa, (**B**) Ciliophora, (**C**) Amoebozoa, and (**D**) Discoba and ‘other phagotrophic protists’. Visualization of the non-random positive (aggregate, blue solid lines) and negative (segregate, orange dashed lines) interactions resulting from pairwise co-occurrence analysis. Line length adjusted to avoid overlap of OTUs. ‘Other phagotrophic protists’ include those phagotrophic protists that do not fall under Ciliophora, Amoebozoa, Discoba or Cercozoa (e.g., Apusomonadida, Telonemia, and phagotrophic Stramenopiles).

**Figure 3 microorganisms-09-01555-f003:**
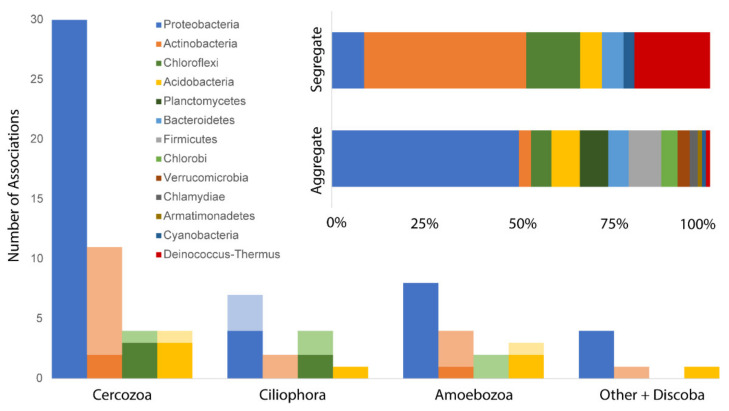
Phagotrophic protists aggregate with Proteobacteria and segregate from Actinobacteria. Bacteria are color-coded by phylum but summed along the *y*-axis as numbers of families (1:1 ratio to associations) per phylum. Bar charts (inset) show proportion of bacterial phyla that aggregate or segregate with all phagotrophic protist response groups combined. Stacked bar graphs (*x*-axis) show proportion of bacterial phyla per phagotrophic protist response group (e.g., Cercozoa). Darker shades (*x*-axis) indicate aggregate associations and lighter shades indicate segregate associations.

**Figure 4 microorganisms-09-01555-f004:**
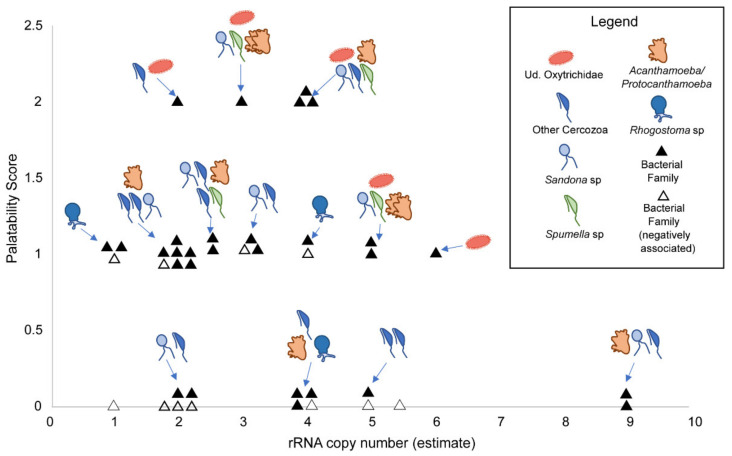
MDV Phagotrophic protists exhibit preferential grazing. A palatability score was assigned to each bacterial family by giving one point for each of the following characteristics: (1) Gram-negative, (2) phototroph, and (3) diazotroph. Associated protists from the network analyses were added and arrows indicate which bacterial cluster the protists are associated with. Roughly half (34 of 74) of the bacterial families were excluded from the analysis because rRNA copy number could not be estimated at the family level. Black-filled triangles denote bacterial families positively associated with a protist in the network analyses, white-filled triangles represented bacterial families negatively associated with a protist in the network analysis. Arrows point to clusters, not individual triangles; each protist can be assumed to be associated with at least one of the bacterial families in a cluster, but how many or which is not specified in this figure.

## Data Availability

Not applicable.

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
