# Peer review of "Phagotrophic Protists and Their Associates: Evidence for Preferential Grazing in an Abiotically Driven Soil Ecosystem"

_microorganisms, 2021, doi:10.3390/microorganisms9081555_

Round 1

Reviewer 1 Report

The paper by Thompson and colleagues analyzed the interactions between phagotrophic protists and other taxonomic groups in the MDV, through co-occurrence among shotgun metagenome sequences, in order to shed light on soil food-webs of such peculiar ecosystems. They found that grazing preferences were mainly determined by the cell-wall structure of prey bacteria. They also addressed the effect of both abiotic and biotic factors on the distribution of phagotrophic protists and on their interactions and they pointed out a predominant effect of biotic variables.

The topic presented is of high interest in the field. In general the paper is of very high quality and well written. Introduction gives a detailed presentation of the knowledge in the field. Method are up to date and presented with high accuracy. The results are presented in detail and with their discussion they represent an advance in the field.

For these reasons, I recommend its acceptance of the publication in the present form.

Author Response

We would like to thank reviewer #1 for their time in reviewing our manuscript.

Reviewer 2 Report

The manuscript by Thompson et al. tried to reveal the interactions of phagotrophic protists with other organisms in the McMurdo Dry Valleys (MDV) of Antarctica. This manuscript was well prepared. The methods were appropriate, the results, especially the interpretation of the network interactions were of great caution. I like this work and believe the manuscript will contribute to the background information of the soil trophic interactions in cold oligotrophic regions like MDV. I have only some minor concerns, and hope these will help to improve the manuscript.

Page 3, line 102. I used to isolate and identify more than 40 cyrtophorian ciliates (including five Pseudochilodonopsis species), but I found none of these three citations here mentioned this genus. But please check and give the proper citations.

Page 4, line 154. Delete either “from” or “across”.

Page 4, lines 181, 182. Stramenopiles appears twice. Please check.

Page 4, lines 190-194. This part is more like a discussion than method.

Page 4, line 195. For the construction of the network, more methodological details have to be shown. For instance, what the input data type, an absolute abundance matrix or a presence-absence one? Which correlation index was adopted, spearman or pearson’s? Your samples were temporal or spatial?

Page 4, lines 199-204. Somehow, I found the definition of being positive and negative is different from “common sense”. Normally, two OTUs are positively related if they have above 0 correlation (>0), while negatively when they have minus correlation (<0). Moreover, thresholds should be determined for the significant positive and negative correlations (using non-model is a good idea). It seems that the authors used the non-model as a standard of being positive or negative, which confuses me. Maybe my understanding is wrong. Please check this.

Page 5, line 236. The collinearity among abiotic factors were tested, then how about the collinearity among abiotic and biotic factors?

Page 6, lines 264-270. I appreciate a lot that the authors interpreted the network results with great caution.

Page 6, line 274. One more possibility should be added: “3) its prey was not included in the network”.

Page 8, line 340. Please briefly describe the full network like how many nodes (OTUs) involved, how many links (interactions) those nodes formed. From this matter, the importance of each group will be shown. Then zoom into the phagotrophic protists.

Page 9. The layout needs to be improved to include the four graphs into one.

Page 15. Also, I think a small section “methodological consideration” should be added to discuss the impact of the method you employed. For instance, the clustering of OTUs into family or genus will reduce the number of the nodes in the network, which will largely reduce the interaction numbers and give less possible trophic indications.

Page 16, The reference form can be improved. Some small issues were found.

1) The article titles in lower case or capital, please keep the consistency.

2) Line 596. The citation is not complete.

3) To list all the authors, or just the first several and then use “et al.”, please check and correct throughout the manuscript.

4) Line 648. The journal abbreviation should be “J Stat Softw”.

Author Response

Page 3, line 102. I used to isolate and identify more than 40 cyrtophorian ciliates (including five Pseudochilodonopsis species), but I found none of these three citations here mentioned this genus. But please check and give the proper citations.

Thompson 2020 mentions the presence of a putative Pseudochilodonopsis sp. in their metagenomes (see supplementals of that paper), but an additional citation was added here to support the claim that the genus could be predatory/cytophagous in the MDV (see Hamels et al. 2004, L102).

Page 4, line 154. Delete either “from” or “across”.

The phrase “from across” is not grammatically incorrect and conveys a different meaning than either “from” or “across” so we have decided to keep it the way it was originally written.

Page 4, lines 181, 182. Stramenopiles appears twice. Please check.

The second reference to “Stramenopiles” was removed. After careful consideration, the entire sentence was removed as we felt it did not add to the section and was not necessary for the reader to know to understand or repeat our study.

Page 4, lines 190-194. This part is more like a discussion than method.

We understand the reviewer’s concern and appreciate their comment. The section is meant to explain why we measured those abiotic variables specifically, especially because several of them are meaningful only in context of the MDV. We have added a short phrase to the beginning of line 190 (i.e., “We measured these attributes specifically because …”

Page 4, line 195. For the construction of the network, more methodological details have to be shown. For instance, what the input data type, an absolute abundance matrix or a presence-absence one? Which correlation index was adopted, spearman or pearson’s? Your samples were temporal or spatial?

We moved the sentence originally at line 204 “To test for associations, normalized counts …” to the beginning of the methods subheading “2.2 Constructing association networks” for clarity. This line explains the input data type, “presence-absence”. A Pearson correlation was used to detect collinearity between abiotic variables (orig. line 237), but neither was used for construction of the network itself because we used a R script that evaluated the probability of co-occurrence and not a correlation to establish likelihood of non-random co-occurrence. We added “spatially associated” to line 128; “… derived from 18 spatially-sampled shotgun metagenomes …”

Page 4, lines 199-204. Somehow, I found the definition of being positive and negative is different from “common sense”. Normally, two OTUs are positively related if they have above 0 correlation (>0), while negatively when they have minus correlation (<0). Moreover, thresholds should be determined for the significant positive and negative correlations (using non-model is a good idea). It seems that the authors used the non-model as a standard of being positive or negative, which confuses me. Maybe my understanding is wrong. Please check this.

The understanding of the reviewer is correct in that we do use a non model in the sense that we compare each pair association against a random probabilistic model of pair association. The random model is built by calculating the probability that a selected pair of species co-occurs at either less or greater frequency than the observed. These probabilities are calculated under the condition that a species probability of occurrence at each site is equal to its observed frequency among all sites and by using combinatorics to determine the number of sites species could co-occur, then if an association is found to be above that random association then it is considered positive and below it is consider negative. Given that the random model is based on the number of sites for each pair the probability can be different for each association, but for all of them it represents the same thing. This method is fully described in Veech, 2013.

Page 5, line 236. The collinearity among abiotic factors were tested, then how about the collinearity among abiotic and biotic factors?

We did not test the collinearity between biotic and abiotic factors a priori. In a sense the shared portion of the variation will indicate this collinearity and is of interest to our study as it can tell us how a portion of the co-occurrence can be explained by both abiotic and biotic factors. We tested the collinearity among abiotic factor to remove redundant factors that could potentially inflate variation being explained and as such could be misleading.

Page 6, lines 264-270. I appreciate a lot that the authors interpreted the network results with great caution.

Page 6, line 274. One more possibility should be added: “3) its prey was not included in the network”.

This is a great suggestion; we have added it to the list (lines 273-274)

Page 8, line 340. Please briefly describe the full network like how many nodes (OTUs) involved, how many links (interactions) those nodes formed. From this matter, the importance of each group will be shown. Then zoom into the phagotrophic protists.

We like the suggestion the reviewer makes about describing the number of nodes and links in each network, however, after reviewing the sections in the results that already deal with this topic we are confident that we have already adequately addressed the number of nodes and links in our networks (lines 343-351 and 369-372).

Page 9. The layout needs to be improved to include the four graphs into one.

This is a good suggestion. We have condensed each of the four network graphs into one larger graph that fits on a single page. We have also modified the figure caption accordingly.

Page 15. Also, I think a small section “methodological consideration” should be added to discuss the impact of the method you employed. For instance, the clustering of OTUs into family or genus will reduce the number of the nodes in the network, which will largely reduce the interaction numbers and give less possible trophic indications.

We agree that highlighting these constraints and their impact on our results is important. To avoid disrupting the flow of the current discussion by adding a new section, we have simply improved on the sections of the manuscript that already discuss these constraints; see lines 172-178, 258-263, and 468-473.

Page 16, The reference form can be improved. Some small issues were found.

  • The article titles in lower case or capital, please keep the consistency.

Capitalization changed to lower case throughout except at beginning of title or after a colon in the title.

  • Line 596. The citation is not complete.

The rest of the citation has been added.

  • To list all the authors, or just the first several and then use “et al.”, please check and correct throughout the manuscript.

We show the first 8 authors of a publication, and if there are more than 8 then we list the first 6 authors, et al., then the final author’s name. We have checked throughout to make sure each reference complied with this format.

4) Line 648. The journal abbreviation should be “J Stat Softw”.

We have edited this reference, and double checked the rest for proper journal abbreviations